# Spatial Evolution of Energetic Electrons Affecting the Upper Atmosphere during the Last Two Solar Cycles

**Alexei V. Dmitriev** [1,2,*], **Alla V. Suvorova** [2], **Sayantan Ghosh** [1], **Gennady V. Golubkov** [3,4] **and Maxim G. Golubkov** [3]

1   Department of Space Sciences and Engineering, National Central University, Taoyuan City 32001, Taiwan; sayantang01@gmail.com
2   Skobeltsyn Institute of Nuclear Physics, Lomonosov Moscow State University, 119991 Moscow, Russia; alla_suvorova@mail.ru
3   Semenov Federal Research Center for Chemical Physics, Russian Academy of Sciences, 119334 Moscow, Russia; kolupanovo@gmail.com (G.V.G.); golubkov@chph.ras.ru (M.G.G.)
4   National Research Center "Kurchatov Institute", 123182 Moscow, Russia
*   Correspondence: dalex@jupiter.ss.ncu.edu.tw; Tel.: +886-03-4228374

**Abstract:** Future commercial, scientific, and other satellite missions require low-Earth-orbit (LEO) altitudes of 300–400 km for long-term successful space operations. The Earth's radiation belt (ERB) is an inevitable obstacle for manned and other space missions. Precipitation of >30 keV energetic electrons from the ERB is one of the sources of ionization in LEO, space vehicles, in the ionosphere, and in the upper atmosphere. We show, in this work, that the area of electron precipitation from the outer ERB shifts equator-wards to Siberia. We further show a substantive decrease in the intensity of energetic electrons in the area of the South Atlantic Anomaly (SAA) from the 23rd to the 24th solar cycles. These results can be attributed to, and explained by, variations in geomagnetic activity, with a noticeable change in the configuration of the Earth's magnetic field during the 24th solar cycle. The diminishing SAA area and electron fluxes should allow elevation of the International Space Station to higher altitudes, thereby making these altitudes accessible to relevant space missions.

**Keywords:** ionosphere; Earth's radiation belt; solar cycle

**PACS:** 94.30.Xy; 94.05.Sd; 94.30.Hn

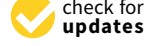



## 1. Introduction

The upper atmosphere is affected by energetic particles precipitating from the Earth's radiation belt (ERB). The ERB is formed by energetic electrons and ions with energies of tens of 30 keV and more. These particles can penetrate to low altitudes and lead to the ionization and excitation of neutral atoms and molecules of the upper atmosphere [1]. Fast electrons with energies E > 30 keV can enter the upper atmosphere at altitudes down to 50 km and, thus, they can ionize the D-, E-, and F-layers of the ionosphere [2]. The precipitation of electrons is accompanied by heating of the upper atmosphere [3], which results in its rise and a change in the chemical composition of the thermosphere [4].

The electrons' ERB consists of an outer and an inner zone. The outer ERB contains fast and relativistic electrons with energies from tens of keV to several MeV, trapped in the magnetosphere [5,6]. The electrons of the outer ERB drift around the Earth at equatorial distances exceeding three Earth radii and oscillate along the lines of force of the geomagnetic field. Most of the trapped electrons have pitch angles near 90° and oscillate in the close vicinity of the geomagnetic equator. Electrons with small pitch angles can precipitate into the ionosphere and the upper atmosphere at mid- and high latitudes, which leads to heating of the atmosphere. In turn, heating of the atmosphere at high latitudes forms neutral winds directed towards the equator, which carry ionospheric and thermospheric disturbances to

lower latitudes [7]. As a result, the precipitation of electrons at high latitudes affects the entire ionosphere, including middle and equatorial latitudes.

The main source of energetic electrons in the outer ERB is the acceleration of the hot plasma of the magnetospheric tail during substorms [8]. Long-lasting periodic magnetic storms are associated with high-speed streams of the solar wind, which are characterized by Alfvén waves of large amplitude in the interplanetary magnetic field [9]. They are accompanied by continuous substorm activity, electron acceleration, and precipitation, which have a constant external effect on the atmosphere, thermosphere, and ionosphere [10]. The total number and power of magnetic storms, as well as the electron fluxes of the outer ERB, are higher in more active solar cycles.

The electrons in the inner ERB have energies of tens to hundreds of keV. At the equator, the electron belt extends from 1.2 to 2.5 Earth radii. Due to the inclination of the axis of the Earth's dipole and its shift relative to the axis of the Earth, the inner ERB falls to low heights in the area of South Atlantic Anomaly (SAA), where intense fluxes of charged particles are constantly recorded. The dipole shift following the model of Thébault et al. [11] continuously decreases, yielding the obvious observation of an average 0.2° shift rate of the northern magnetic dip-pole moving with an acceleration from Canada towards Siberia in the last two decades. At the altitudes of the inner ERB, the magnetic field increases, resulting in a gradual decrease in particle fluxes and SAA area [11]. Additionally, the SAA area is affected by the particle energy losses due to ionization in the atmosphere. It has been established in [12] that, during high solar and geomagnetic activity, the atmosphere heats up and rises, which leads to an increase in losses and a decrease in the SAA area, in contrast to low solar activity periods, when this area increases.

At low latitudes and heights of several hundred kilometers, i.e., under the inner ERB, sporadic intense fluxes of electrons with energies of $10\,\text{keV} < \text{E} < 300\,\text{keV}$ are also observed [13,14]. In this region, electrons drift across the geomagnetic field in longitude to the east and in less than 20 h reach the SAA, where they descend to heights below 100 km, which results in their loss in the atmosphere due to ionization. Such electrons are called quasi-trapped, since their lifetime is limited to only one revolution around the Earth. It was shown by Suvorova et al. [15] that the source of these electrons is the inner ERB. Thus, energetic electrons at low latitudes are able to penetrate into the ionosphere and upper atmosphere at practically any longitude, not only in the SAA region. However, the mechanism of their transport from the ERB has not been fully investigated.

We develop a novel statistical method for analyzing the dynamics of charged ERB particles presented in our previous works [1,16]. The aim of the present work is to determine the spatial distribution and evolution of the precipitation of fast electrons from the ERB into the upper atmosphere by using continuous low-orbit NOAA/POES satellite observations during solar cycles 23 and 24 (i.e., from 1998 to 2019), which provide the largest statistics available up to the present day. In addition to the dynamics of electron precipitations from the outer ERB, we investigate the long-term dynamics of electrons penetrating from the inner ERB. We search for signatures in the data on ERB during the two solar cycles as a function of the variations in the geomagnetic activity and the configuration of the Earth's magnetic field.

## 2. Experimental Data

Fluxes of energetic electrons are continuously observed by low-orbit polar satellites of the NOAA/POES series from the year of 1998 [17]. The satellites have a sun-synchronous orbit with an inclination of 98° at an altitude of approximately 850 km. Three pairs of satellites moving in three orbital planes cover the ranges of terminators at 06:00 and 18:00 LT, morning–evening at 09:00 and 21:00 LT, and day–night at 02:00 and 14:00 LT. This makes it possible to conduct almost simultaneous measurements in different regions of the magnetosphere. The NOAA-15 probe provided data from 1998 to 2019; NOAA-16 from 2001 to 2014; NOAA-17 from 2002 to 2013; NOAA-18 from 2005 to 2019; NOAA-19

from 2009 to 2019; METOP-1 from 2014 to 2019; and METOP-2 from 2006 to 2019. We have summarized the data description in Table 1.

**Table 1.** Data availability of NOAA/POES satellites.

| Year | Probe | LT Range | Year | Probe | LT Range |
|------|-------|----------|------|-------|----------|
| 2001 | NOAA-15 | 06:00 & 18:00 | 2013 | NOAA-15 | 06:00 & 18:00 |
|      | NOAA-16 | 06:00 & 18:00 |      | NOAA-16 | 06:00 & 18:00 |
|      |         |              |      | NOAA-18 | 02:00 & 14:00 |
|      |         |              |      | NOAA-19 | 02:00 & 14:00 |
|      |         |              |      | METOP-2 | 09:00 & 21:00 |
| 2004 | NOAA-15 | 06:00 & 18:00 | 2016 | NOAA-15 | 06:00 & 18:00 |
|      | NOAA-16 | 06:00 & 18:00 |      | NOAA-18 | 02:00 & 14:00 |
|      | NOAA-17 | 09:00 & 21:00 |      | NOAA-19 | 02:00 & 14:00 |
|      |         |              |      | METOP-1 | 09:00 & 21:00 |
|      |         |              |      | METOP-2 | 09:00 & 21:00 |
| 2007 | NOAA-15 | 06:00 & 18:00 | 2019 | NOAA-15 | 06:00 & 18:00 |
|      | NOAA-16 | 06:00 & 18:00 |      | METOP-1 | 09:00 & 21:00 |
|      | NOAA-17 | 09:00 & 21:00 |      | METOP-2 | 09:00 & 21:00 |
|      | METOP-2 | 09:00 & 21:00 |      | NOAA-18 | 02:00 & 14:00 |
|      | NOAA-18 | 02:00 & 14:00 |      | NOAA-19 | 02:00 & 14:00 |

Figure 1 shows the solar activity and geomagnetic activity represented, respectively, by sunspot numbers and planetary daily $A_P$ index during the years from 1998 to 2019. It can be seen that this time interval covers almost two solar cycles: the 23rd solar cycle (from 1998 to 2009) and the 24th solar cycle (from 2010 to 2019). Solar maxima occurred, respectively, in 2000–2001 and 2012–2013. For the comparative analysis at different phases of the solar cycle, we selected a year from each solar cycle: the solar maximum (2001 and 2013), declining phase (2004 and 2016), and immediately before the solar minimum (2007 and 2019). As one can see in Figure 1, the 24th cycle is weaker in solar and geomagnetic activity than the 23rd one. The geomagnetic disturbances do not follow directly the solar activity. Namely, more disturbed days occur during the declining phase. The geomagnetic activity is minimal during the solar minima.

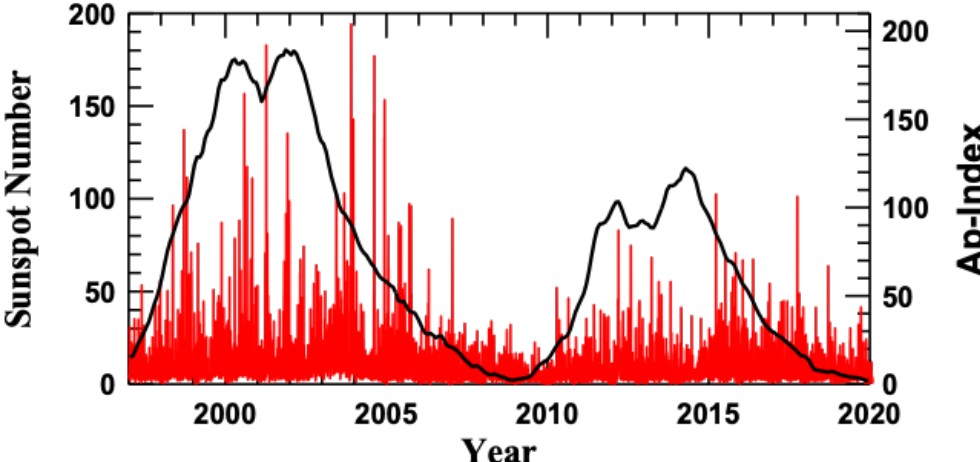

**Figure 1.** Solar activity represented in sunspot number (left axis) and geomagnetic activity represented by Ap index (right axis) during the 23 and 24 solar cycles.

The data availability from NOAA/POES satellites during the years selected is presented in Table 1. It can be seen that the year of 2001 is poor in statistics, both in the number of satellites and in the local time coverage relative to other years. From the year of 2007,

five POES probes cover all the three LT ranges. Hence, the comparison of effects during these years will not be affected by the statistics.

We used data provided by the Medium Energy Proton and Electron Detector (MEPED) onboard the POES satellites. MEPED includes two identical solid-state detector telescopes, which measure particle fluxes in horizontal and vertical directions, respectively (https://satdat.ngdc.noaa.gov, accessed on 31 December 2021). We analyzed measurements of >30 keV electrons observed by a zenith-directed detector. At low latitudes, the magnetic field lines are located almost horizontally. Hence, the detector observes quasi-trapped particles, which quickly lose the energy in the upper atmosphere. Namely, during the eastward azimuthal drift, the electron drift shells descend to a dense atmosphere in the SAA region. At high latitudes, where the field lines are almost vertical, the detector observes electrons precipitating from the outer ERB into the loss cone and penetrating to the heights of the upper atmosphere.

The numerical analysis of electron precipitations was conducted following Suvorova and Dmitriev [18] and Suvorova [19], where we split the geographic map into spatial cells of $3° \times 2°$ resolution (for latitude and longitude, respectively). For each cell, we determined the maximum electron flux from all available POES satellite data during the given year. The resulting annual geographic maps were further used to calculate the normalized number of occurrences of maximal electron fluxes $I > 1 \, \text{cm}^{-2}\text{s}^{-1}\text{sr}^{-1}$.

We should note that, until 2014, satellite data were acquired with a temporal resolution of 16 s and afterwards with a resolution of $\sim$2 s. We remind the reader that the number of satellites was increased after 2014, with the successor program METOP, thereby increasing the number of observations by $\sim$10 times. This represents the most complete set of energetic electron flux measurements in the low Earth orbit to date. The increased number of observation platforms (from two to six by 2009) and the relatively large spatial scale of $3° \times 2°$, however, has a negligible effect on our study, since, except for 2001, the other years are well represented in the data and, thus, are directly comparable.

## 3. Results

We present the geographic maps of electron fluxes $> 30$ keV measured by NOAA/POES satellites during different phases of two solar cycles in Figure 2. At high and middle latitudes above 40° in the northern and southern hemispheres, one can observe wide streaks of precipitation from the outer ERB with intensities $I > 1 \times 10^7 \, \text{cm}^{-2}\text{s}^{-1}\text{sr}^{-1}$. At low latitudes, fluxes of quasi-trapped electrons from the inner ERB are readily observable and stretch along the geomagnetic equator, with a maximum in the region of SAA at longitudes approximately from $-100°$ to $-20°$.

A comparison of the electron precipitations from the outer ERB shows that the intensity of such precipitations was higher in solar cycle 24, as compared to solar cycle 23. Specifically, 2004 and 2016 (during the declining phase of the cycle) showed the most intense precipitations. We additionally observed that intense precipitations at eastern longitudes (around 100° E) in the northern hemisphere appeared at lower latitudes (with an equator-ward shift of several degrees) in the 24th solar cycle as compared to the previous cycle. This shift was more prominent in 2019, confirming earlier findings [1,20,21], and is related to the accelerated motion of the north magnetic dip-pole from Canada towards Siberia [11] (see the bottom panel of Figure 2).

However, at lower latitudes, the observed effects exhibit a contrasting picture. Namely, the fluxes of quasi-trapped electrons in the 24th cycle are noticeably weaker than in the 23rd cycle, when solar activity and the number of powerful magnetic storms were much higher. The most intense fluxes for a given solar cycle were observed during the declining phase. We show in Figure 2 that the dynamics of quasi-trapped electrons can be traced and, thus, can be used to determine the mechanism of their appearance in the forbidden region under the inner ERB zone. We distinguish two areas of intense fluxes, the first of which is located over Africa and extends from the eastern edge of the SAA to longitudes of $\sim$60° E, while the second area is located over the Pacific Ocean and extends from longitudes of

~100° E to the western boundary of the SAA. Between these areas, there is a gap, where the electron fluxes are rather weak and rare. Note that the fluxes of quasi-trapped electrons over Africa are much weaker than over the Pacific Ocean. The fluxes of electrons over the Pacific Ocean increase with longitude, reaching a maximum in the SAA region.

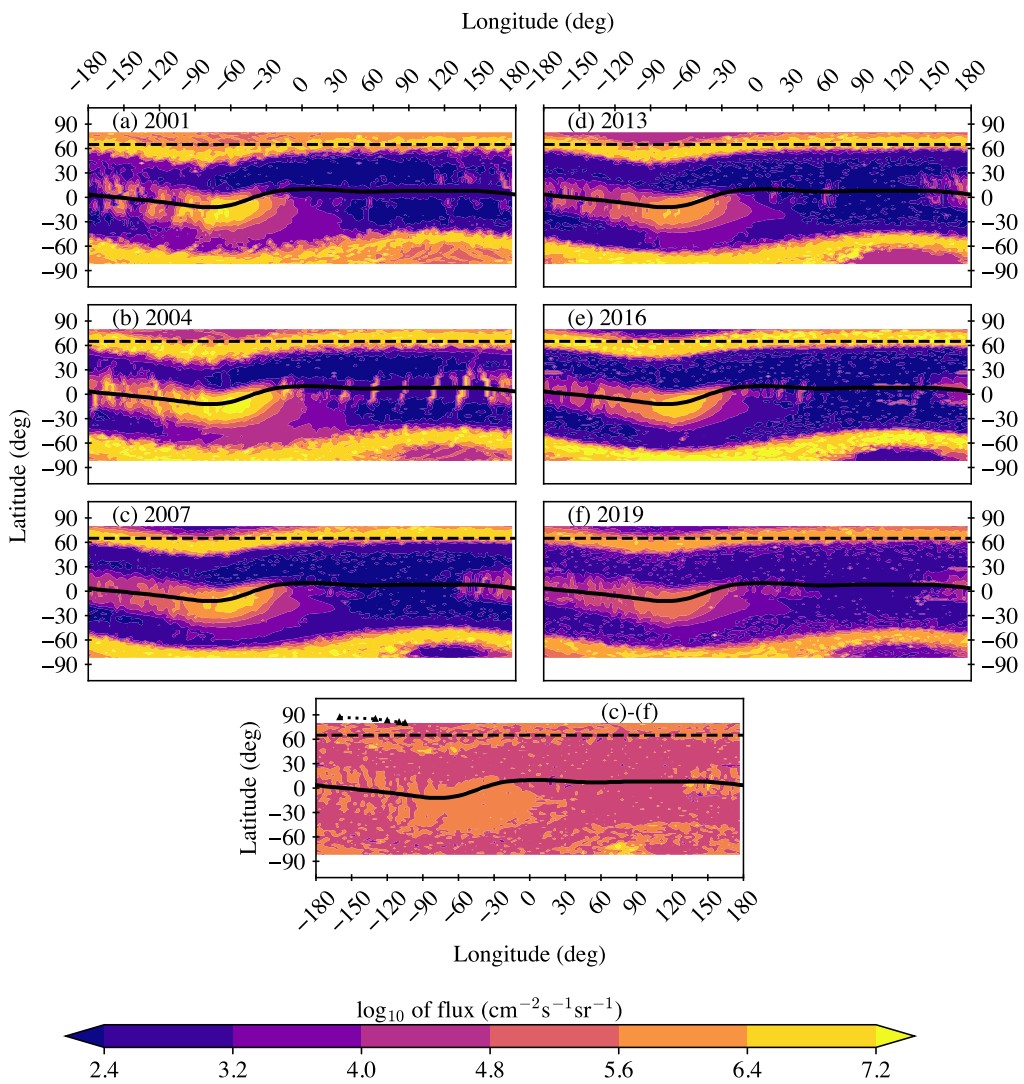

**Figure 2.** Geographic maps of >30 keV electron fluxes measured by NOAA/POES satellites during different levels of the solar activity in the 23rd solar cycle (left column) and in the 24th solar cycle (right column): upper panels—solar maximum (the years of 2001 and 2013, respectively); middle panels—declining phase (the years of 2004 and 2016, respectively); and near the solar minimum (the years of 2007 and 2019, respectively). Geomagnetic equator is shown by the black curve. The horizontal dashed lines depict the latitude of 65°. The bottom panel represents the difference in the electron flux maps between 2019 and 2007. The trace of the northern magnetic dip-pole from 1995 to 2015 is indicated with a black dotted line with triangular nodes at every five years (from right to left) in the bottom panel.

We can observe a significant temporal decrease in the >30 keV electron flux intensity in the SAA region in Figure 2, with the last solar minimum displaying the minimum flux around 2019. This can be related to a decrease in the SAA area. Figure 3 shows the probability of occurrence of intense >30 keV electron fluxes with $I > 1 \times 10^4 \, \text{cm}^{-2}\text{s}^{-1}\text{sr}^{-1}$ during different phases of the 23rd and 24th solar cycles. The different rate and volume of observations in the different years, as mentioned in the last section, warranted a nor-

malization of the occurrences by the total number of observations in a given year, thereby yielding a probability of occurrence that can be reliably compared.

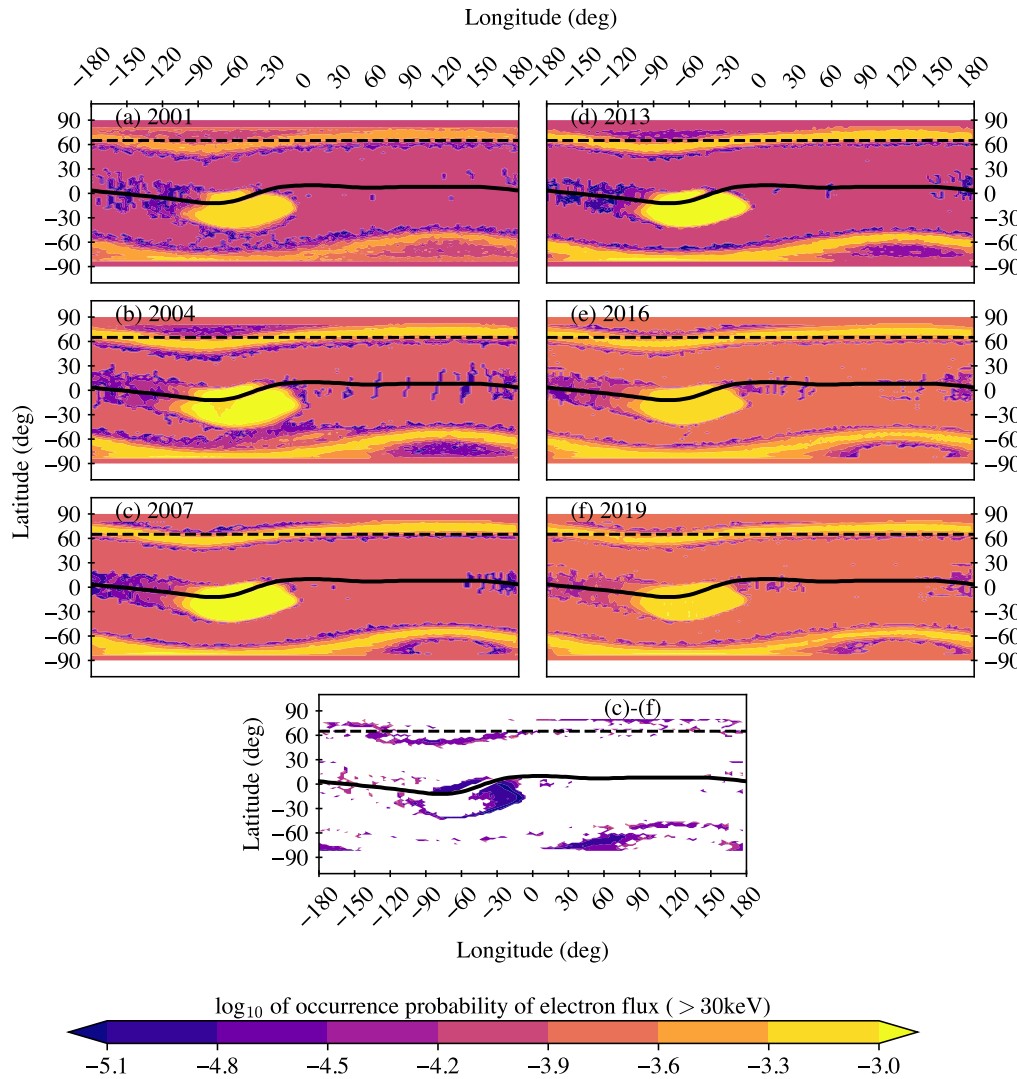

**Figure 3.** Geographic maps of normalized occurrence probability to detect $>30\,keV$ electron fluxes with intensity $> 1 \times 10^4\,cm^{-2}s^{-1}sr^{-1}$ within one year during different levels of solar activity in the 23rd solar cycle (left column) and in the 24th solar cycle (right column): upper panels—solar maximum (the years of 2001 and 2013, respectively); middle panels—declining phase (the years of 2004 and 2016, respectively); and near the solar minimum (the years of 2007 and 2019, respectively). Geomagnetic equator is shown by the black curve. The bottom panel represents the difference between 2007 and 2019.

Figure 3 shows that the low-latitude region of intense electron fluxes observed in the 23rd cycle has a significantly greater extent both in latitude and longitude than that observed in the 24th cycle. The westward-moving (from SAA) plume of intense flux along the geomagnetic equator can be seen to have a noticeably higher latitudinal extent in the 23rd solar cycle. The largest SAA area was observed in 2004, during the declining phase of solar cycle 23. It is, however, important to note the continuous decrease in the SAA area during the 24th solar cycle, reaching its minimum in 2019.

## 4. Discussion

To summarize the analysis discussed above, in this work, we demonstrate a shift in the E $>$ 30 keV electron precipitations over eastern longitudes (Siberia) towards the

equator in the 24th solar cycle, thus confirming the discoveries in Dmitriev [20,21]. These current results present an important step towards understanding the dynamics of the equator-ward shift in intense electron precipitation during different phases of the solar cycle, and also lay out a possible direction for understanding the processes involved in the transport of electrons from the outer ERB at the middle and high latitudes of the northern hemisphere. The accelerated shift of the northern magnetic pole from Canada towards Siberia [11] possibly plays a significant role in these observed effects, and they are affected by geomagnetic jerks, which are sudden changes in the acceleration of the geomagnetic field [20].

Our observations in this work suggest a larger shift during the 24th solar cycle than would be expected from the IGRF model [21], resulting in the observation of discrete auroras in mid-latitude Russia accompanied by intense electron precipitations from the outer ERB. An increased solar wind–magnetosphere coupling might provide an additional explanation, which in turn might be caused by a higher efficiency of interaction with high-speed solar winds, and this requires further sophisticated investigations.

At the same time, the significant decrease in the intensity of electron flux observed at low latitudes, including the SAA region, accompanied by a continuously decreasing SAA area during the two solar cycles cannot be sufficiently explained by decreasing solar activity, which would have caused the SAA to increase [12]. Since the 24th solar cycle was anomalously weak compared to previous cycles (see Figure 1), a noticeable weakening of the sources and fluxes of the ERB electrons can be posited. However, although the outer ERB exhibits an increase in the electron fluxes, the inner ERB shows a corresponding decrease. This suggests that the outer ERB should not be considered as a source for the inner ERB (as has been claimed in the literature). A further effect is the accelerated change in the geomagnetic field, resulting in increased magnetic field strength in SAA region. This increase results in a rise in the lower edge of the inner ERB to greater heights, supplemented by a decrease in the flux of trapped particles [11].

The analysis of the distribution of quasi-trapped electrons at low latitudes made it possible to determine the mechanism of their appearance under the inner ERB at an altitude of 850 km. This mechanism is directly related to the topology of the Earth's magnetic field, which has a minimum strength in the SAA region at a given altitude in the geomagnetic equator. The magnetic field grows rapidly to the east and reaches a maximum in the region of longitudes from 60° E to 100° E. Further, the magnetic field begins to decrease gradually and returns to its minimum values in SAA. Those electrons that are beginning their movement from the SAA region drift eastward along the drift shells and, thus, they quickly move upward to higher heights to the region of lower magnetic field strength, while maintaining the magnetic moment. Having passed the region of the minimum field value, the electrons begin to move downward and become visible again at a given altitude. In this case, the flow of electrons should not increase. The increase in the flux and frequency of observation of energetic electrons over the Pacific Ocean can be explained by the injection of particles from the inner ERB. The distributions obtained in this work are a reliable confirmation of the mechanism of anomalous radial transport of energetic electrons from the inner ERB toward the Earth, followed by an azimuthal drift to the east [22]. This mechanism is also supported by the noticeable decrease in the quasi-trapped electron fluxes, which can also be explained by the elevation of the inner ERB in the SAA region.

## 5. Conclusions

We have presented an analysis performed on continuous observations covering the 23rd and 24th solar cycles (i.e., from 1998 to 2019) recorded by detectors onboard the low-Earth-orbit NOAA/POES polar satellites. We analyzed the spatial distributions of E > 30 keV energetic electrons precipitating from the ERB to the upper atmosphere. The comparative analysis of satellite observations shows the following:

- The area of electron precipitation from the outer ERB shifted over eastern longitudes (Siberia) toward the equator. This is in good agreement with the latest data on the dynamics of the geomagnetic North Pole.
- The intensity of energetic electron fluxes in the SAA region, as well as its area, significantly decreased in the 24th solar cycle compared to the 23rd cycle. This was caused mainly by prominent changes in the geomagnetic field at low latitudes such that the magnetic field strength increased in the SAA region, which resulted in elevation of the inner ERB.
- The distribution of quasi-trapped electrons obtained in this work at low latitudes makes it possible to confirm the mechanism of their appearance under the inner ERB zone at an altitude of 850 km due to anomalous earthward radial transport from the inner ERB zone. The latter is especially important to take into account in order to increase the stability of the operation of global navigation satellite systems and remote sensing of the Earth [23–25].
- In order to separate the effects of geomagnetic activity and the geomagnetic field changes in the electron ERB dynamics, we plan additional sophisticated analyses with more data in the near future.

Finally, the diminishing SAA area and fluxes make the elevation of the International Space Station to higher altitudes possible, thereby making new altitudes (in the range of 300–400 km) accessible to satellites relevant for commercial and scientific purposes.

**Author Contributions:** Conceptualization, G.V.G.; Methodology, M.G.G.; Software, A.V.D.; Formal analysis, A.V.S.; Writing—original draft preparation, A.V.D. and S.G.; Writing—review and editing, M.G.G. and A.V.S.; Visualization, A.V.D. and S.G.; Supervision, G.V.G.; Project administration, A.V.D. All authors have read and agreed to the published version of the manuscript.

**Funding:** The study was carried out as part of a state assignment of the Ministry of Science and Higher Education of the Russian Federation (theme no. 1021051201992-1). The work of A.V.D and A.V.S. was supported by the MOST (grant nos. 108-2811-M-008-518 and 108-2111-M-008-035, respectively) as well as by the Research Foundation of the National Central University. S.G. was supported by MOST, Taiwan (grant nos. MOST 110-2111-M-008-034-MY2 and MOST 110-2111-M-008-013).

**Institutional Review Board Statement:** Not applicable.

**Informed Consent Statement:** Not applicable.

**Data Availability Statement:** All the data used in this study are publicly available. The NOAA/POES data were obtained from https://satdat.ngdc.noaa.gov, accessed on 31 December 2021. The solar activity and geomagnetic index were obtained from NASA Goddard Space Flight Center's CDAWeb https://cdaweb.gsfc.nasa.gov, accessed on 31 December 2021.

**Acknowledgments:** A.V.D. and S.G. acknowledge Bhavana Lalchand for the patient and through reading of the manuscript and invaluable help in editing the same.

**Conflicts of Interest:** The authors declare no conflict of interest.

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
