# Peer review of "Spatial Evolution of Energetic Electrons Affecting the Upper Atmosphere during the Last Two Solar Cycles"

_atmosphere, doi:10.3390/atmos13020322_

Round 1

Reviewer 1 Report

Please see attached PDF.

Author Response

We appreciate the Reviewers’ #1 critical comments and suggestions. We have revised the manuscript accordingly.

The Authors analyzed long-time dynamics of electron fluxes with energies > 30 keV at an altitude of 850 km based on the satellite data during 23rd and 24th solar cycles. My opinion is that the authors should provide a more complete analysis that the manuscript be acceptable for publication in the Atmosphere journal. Namely, the authors presented three figures, one of which is the number of sunspots during the 23rd and 24th solar cycles which can be find at many websites.

We have improved Figure 1 and demonstrate both solar and geomagnetic activity for better understanding the main sources of ERB variations. We add a description of this important issue:

“Figure 1 shows the solar activity and geomagnetic activity represented, respectively by sunspot numbers and planetary daily Ap-index during the years from 1998 to 2019. It can be seen that this time interval from 1998 to 2019 covers almost two solar cycles: the 23rd solar cycle (from 1998 to 2009) and the 24th solar cycle (from 2010 to 2019): solar maxima occurred, respectively, in 2000-2001 and 2012-2013. For the comparative analysis at different phases of the solar cycle, we have selected a year from each solar cycle: solar maximum (2001 and 2013), declining phase (2004 and 2016), and right before solar minimum (2007 and 2019). As one can see in Figure 1, the 24th cycle is weaker in solar and geomagnetic activity than the 23rd one. The geomagnetic disturbances do not follow directly the solar activity. Namely, more disturbed days with the daily Ap > 20 occur during the declining phase. The geomagnetic activity is minimal during the solar minima.”

The other two figures refer to satellite data available on the Internet and, as far as I understand, no data processing has been done. Some of the conclusions were not accompanied by adequate analysis, and because of all that, my recommendation is major revision which should include answers/comments to the following remarks.

  1. The authors should clearly state what is new in this paper. Is this a methodology that has shown that the area of electron precipitation from the outer ERB shifts equatorward over Siberia?

We clarify this issue in the end of Introduction:

“We develop a novel statistical method for analysing the dynamics of charged ERB particles presented in our previous works [1,16]. The aim of the present work is to determine the spatial distribution and evolution of the precipitation of fast electrons from the ERB into the upper atmosphere by using continuous low orbit NOAA/POES satellite observations during solar cycles 23 and 24 (i.e., from 1998 to 2019), which provide the largest statistics available up to the present day. In addition to the dynamics of electron precipitations from the outer ERB, we investigate the long term dynamics of electrons penetrating from the inner ERB. We are searching for signatures in the data on ERB during the two solar cycles as a function of the variations in the geomagnetic activity and the configuration of the Earth's magnetic field.”

  1. The authors should explain how they used data from satellites because of three reasons:

2.1) There is a time overlap (in years and in local time (LT)) in the work of several satellites - explain the procedure for obtaining data in this case;

In order to make more clear the statistics both in the number of NOAA/POES probes and in the coverage of LT, we have revised Table 1  where we show the coverage of probes and LT for the years selected. I believe that this helps the reader to understand the whole statistics we use.

We also revised the description of data analysis technique:

“The numerical analysis of electron precipitations was conducted following Suvorova and Dmitriev [18] and Suvorova [19], where, we split the geographic map into spatial cells of 3° × 2° resolution (for latitude and longitude respectively). For each cell, we determined the maximum electron flux from all available POES satellite data during the given year. The resulting annual geographic maps were further used to calculate the normalised number of occurrences of maximal electron fluxes I > 1×104 cm2s1sr1

2) If there is a mechanism for using data from several satellites, is the quality of data worse in the periods of operation of only one satellite (1998-2000)?

The quality of data from one satellite is not worse. But the statistics is suffered from a pure spatial and temporal coverage. This is why we do not use the pair of years 1999 – 2011 for the comparison: the statistics is crucially different. We believe that the NOAA/POES fleet (as well as we:) will survive until the year of 2024. I this case, we will be able making a comparison at the rising phase.

Based on the information presented in the revised version of Table 1, we write the following in the revised text:

“The data availability from NOAA/POES satellites during the years selected is presented in Table 1. It can be seen that the year of 2001 is poor in statistics both in number of satellites and in the local time coverage relative to other years. From the year of 2007, five POES probes cover all the three LT ranges. Hence, the comparison of effects during these years will not be affected by the statistics.”

3) How do the measurements in different LTs fit, first of all how do the measurements related to night time fit with other ones?

From the new Table 1, we can see that in the years 2007 and 2019, all the possible ranges of LT are covered. It should be noted here that the local time, indicated in Table 1, is provided for the equatorial region. At high latitudes, satellites cross practically all MLTs and, thus, for the 1-year data this effect can be neglected. 

We have add an addition paragraph in order to discuss this very important issue:

“We should note that until 2014, satellite data were acquired with a temporal resolution of 16 s, and after that, with a resolution of ~ 2 s. We remind the reader that the number of satellites were increased after 2014, with the successor program METOP, thereby increasing the number of observations by ~ 10 times. This represents the most complete set of energetic electron flux measurements in the low Earth orbit to date. The increased number of observation platforms (from two to six by 2009), and the relatively large spatial scale of 3° x 2° however, has negligible effect on our study; since, except for 2001, the other years are well represented in the data and, thus, are directly comparable.”

  1. Lines 117-118: How are the values -100o and -20o determined?

The location of SAA is determined here approximately as a region of strong electron fluxes coming from the lower edge of the inner ERB.

We have revised the text accordingly:

“At low latitudes, one can see fluxes of quasi-trapped electrons from the inner ERB, which stretches along the geomagnetic equator with a maximum in the region of SAA at longitudes approximately from -100° to -20°.”

  1. Lines 121-125: This statement should be accompanied by a visualization of the corresponding differences. In other words, the authors should figures/panels that would show this shift. This shift should be compared with the corresponding shifts in the literature.

We have modified Figure 2 and show an additional panel, which represents the difference in the

electron flux maps between 2019 and 2007.

In the text, we write the following:

“This shift is more prominent in 2019, confirming earlier findings [1,20,21], and is related to the accelerated motion of the north magnetic dip-pole from Canada towards Siberia [11] (see the bottom panel of Figure 2).”

  1. The authors should clearly state in the text the difference between the values shown in Figs 2 and 3.

In the revised manuscript, we explain the difference in constructing the Figures:

“The numerical analysis of electron precipitations was conducted following Suvorova and Dmitriev [18] and Suvorova [19], where, we split the geographic map into spatial cells of 3°×2°resolution (for latitude and longitude, respectively). For each cell, we determined the maximum electron flux from all available POES satellite data during the given year. The resulting annual geographic maps were further used to calculate the normalised number of occurrences of maximal electron fluxes I > 1×10^4 cm^-2 s^−1 sr^−1.”

  1. Lines 166-168: The accelerated shift of the north magnetic field from Canada towards Siberia should be shown in a figure so that it is clear to readers who are not experts in this research field. The authors should present a comparison of the displacement of the area of precipitations over eastern longitudes and the north magnetic field in the observed time period and give an appropriate analysis.

At the bottom panel in Figure 2, we draw a trace of the Northern magnetic dip pole from 1995 to 2015 in according to [11]. In the revised text, we describe it (see above).

  1. Authors should cite several references of other authors. Currently 50% of references are to the works of the authors of this manuscript.

These new effects have been found recently (in 2019). We do not know other papers related to them. We will very appreciate the reviewer for any relevant paper.

  1. Numbers and axis titles should be larger.

Corrected

Reviewer 2 Report

The Authors analyzed long-time dynamics of electron fluxes with energies > 30 keV at an altitude of 850 km based on the satellite data during 23rd and 24th solar cycles. My opinion is that the authors should provide a more complete analysis that the manuscript be acceptable for publication in the Atmosphere journal. Namely, the authors presented three figures, one of which is the number of sunspots during the 23rd and 24th solar cycles which can be find at many websites. The other two figures refer to satellite data available on the Internet and, as far as I understand, no data processing has been done. Some of the conclusions were not accompanied by adequate analysis, and because of all that, my recommendation is major revision which should include answers/comments to the following remarks.

1. The authors should clearly state what is new in this paper. Is this a methodology that has shown that the area of electron precipitation from the outer ERB shifts equatorward over Siberia?

2. The authors should explain how they used data from satellites because of three reasons: 1) There is a time overlap (in years and in local time (LT)) in the work of several satellites - explain the procedure for obtaining data in this case; 2) If there is a mechanism for using data from several satellites, is the quality of data worse in the periods of operation of only one satellite (1998-2000)? 3) How do the measurements in different LTs fit, first of all how do the measurements related to night time fit with other ones?

3. Lines 117-118: How are the values -100o and -20o determined?

4. Lines 121-125: This statement should be accompanied by a visualization of the corresponding differences. In other words, the authors should figures/panels that would show this shift. This shift should be compared with the corresponding shifts in the literature.

5. The authors should clearly state in the text the difference between the values shown in Figs 2 and 3.

6. Lines 166-168: The accelerated shift of the north magnetic field from Canada towards Siberia should be shown in a figure so that it is clear to readers who are not experts in this research field. The authors should present a comparison of the displacement of the area of precipitations over eastern longitudes and the north magnetic field in the observed time period and give an appropriate analysis.

7. Authors should cite several references of other authors. Currently 50% of references are to the works of the authors of this manuscript.

8. Numbers and axis titles should be larger.

Author Response

We are very grateful to the Reviewer #2 for very valuable comments and useful suggestions. They help us to improve the quality of the paper significantly.

Spatial Evolution of Energetic Electrons Affecting the Upper Atmosphere during the Last Two

Solar Cycles

Alexey V. Dmitriev, Alla V. Suvorova, Gennady V. Golubkov and Maxim G. Golubkov

This is a very interesting paper that presents a analysis performed on electron flux

data (energy > 30 keV, altitude = 850 km) covering two solar cycles. The data are analyzed for

different levels of solar activity (solar maximum, declining phase, near solar minimum) and the two

solar cycles are compared. Observed effects in the data are explained by variations in geomagnetic

activity and by changes in the configuration of Earth's magnetic field. Below, some questions and

comments for the authors, as well as some edits, for their consideration.

Lines 14-16: We analyzed long-time dynamics of electron fluxes with energies > 30

keV at an altitude of 850 km based on the largest statistics to date, which covers two solar cycles 23rd

and 24th.

Covering two solar cycles (23 and 24), we analyzed the long-time

dynamics of electron fluxes with energies > 30 keV, at an altitude of 850 km, based on the

largest statistics to date.

Corrected

Lines 56- The dipole shift continuously decreases, as follows from long-term

observations of the geomagnetic field.

Suggest to add how many degrees per year and to discuss what this potentially would mean

in Section 4 if this trend continues while referring to what was found in the POES data when

comparing the two solar cycles.

The effects related to changes in the geomagnetic field are described in [11].

In Introduction section, we discuss this issue as the following:

“The dipole shift following the model of Thйbault et al. [11] continuously decreases, yielding to the obvious observation of an average 0.2° shift rate of the northern magnetic dip-pole moving with an acceleration from Canada towards Siberia in the last two decades. At the altitudes of the inner ERB, the magnetic field increases resulting in a gradual decrease of particle fluxes, and SAA area [11].” 

In Section 4 we also discuss the effects related to the changes in geomagnetic field as the following:

“To summarise the analysis discussed above, in this work, we demonstrate a shift of the E > 30 keV electron precipitations over eastern longitudes (Siberia) towards the equator in the 24th solar cycle, thus confirming the discoveries in Dmitriev [20,21]. These current results, present an important step towards understanding the dynamics of the equator-ward shift of intense electron precipitation during different phases of the solar cycle, and also outlay a direction for understanding the processes involved in the transport of electrons from the outer ERB at the middle and high latitudes of the northern hemisphere. The accelerated shift of the northern magnetic pole from Canada towards Siberia [11] possibly plays a significant role in these observed effects, and are affected by geomagnetic jerks, which are sudden changes in the acceleration of the geomagnetic field [20].”

We also write in the Conclusions Section the following:

“Finally, the diminishing SAA area and fluxes makes elevation of the International Space Station to higher altitudes possible, thereby making new echelons (in the range of 300km - 400 km) accessible to satellites relevant for commercial, and scientific purposes.”

Lines 61- In contrast, during low solar activity, this are

this area known to increase during low solar activity or is this yet to be proven with data? Suggest to

rephrase sentence with the answer.

The sentences have been rephrased accordingly:

“It has been established in [12] that during high solar and geomagnetic activity, the atmosphere heats up and rises, which leads to an increase in losses and a decrease in the SAA area in contrast to low solar activity periods, when this area increases.”

Lines 74-76: atmosphere using continuous low-orbit satellite observations during the 23rd and 24th

solar cycles (i.e., from 1998 to 2019) that provided the largest statistics available on present day.

atmosphere by using continuous low-orbit NOAA/POES satellite observations during solar cycles

23 and 24 (i.e., from 1998 to 2019) that provide the largest statistics available up to the present day.

Corrected

Line 73-78: Suggest that the paragraph begins by briefly presenting the previous works [1, 16]. Then

continue with The aim of the present work explain how the work presented in this paper

differs (e.g., searching for signatures in the data during the two solar cycles as function of variations

in geomagnetic activity and the configuration of the Earth's magnetic field).

The paragraph has been revised accordingly:

“We develop a novel statistical method for analysing the dynamics of charged ERB particles presented in our previous works [1,16]. The aim of the present work is to determine the spatial distribution and evolution of the precipitation of fast electrons from the ERB into the upper atmosphere by using continuous low orbit NOAA/POES satellite observations during solar cycles 23 and 24 (i.e., from 1998 to 2019), which provide the largest statistics available upto the present day. In addition to the dynamics of electron precipitations from the outer ERB, we investigate the long term dynamics of electrons penetrating from the inner ERB. We are searching for signatures in the data on ERB during the two solar cycles as a function of the variations in the geomagnetic activity and the configuration of the Earth's magnetic field.”

Line 83: Provide some more explanation regarding having Three pairs of satellites

in Table 1.

We add the following sentence:

“This makes it possible to conduct almost simultaneous measurements in different regions of the magnetosphere.”

Line 90: IT IS WRITTEN lost the energy lose their energy

Corrected

Lines 85-87: The satellites are equipped with detectors of electrons and protons

coming from different directions. We used a detector directed towards the zenith, which measure

electrons with energy E> 30 keV (http://satdat.ngdc.noaa.gov).

Lines 95-96: To construct the dataset covering two solar cycles (Table 1) were electron flux data

originating from different satellites cross-calibrated?

The MEPED detectors onboard NOAA/POES satellites from NOAA-15 and after belong to SEM-2 era. In our previous analysis of >30 kev electron data, we did not find significant differences in the electron fluxes measured by different NOAA/POES satellites [Suvorova et al., 2013; 2016].

In the text, we clarify these important issues as the following:

“We used data provided by the Medium Energy Proton and Electron Detector (MEPED) onboard the POES satellites. MEPED includes two identical solid-state detector telescopes, which measure particle fluxes in horizontal and vertical directions respectively (https://satdat.hgdc.noaa.gov). We analysed measurements of > 30 keV electrons observed by zenith-directed detector.”

References:

Suvorova, A. V., A. V. Dmitriev, L.-C. Tsai, V. E. Kunitsyn, E. S. Andreeva, I. A. Nesterov, and L. L. Lazutin (2013), TEC evidence for near-equatorial energy deposition by 30 keV electrons in the topside ionosphere, J. Geophys. Res. Space Physics, 118, 4672-4695, doi:10.1002/jgra.50439.

Suvorova, A. V., C.-M. Huang, A. V. Dmitriev, V. E.Kunitsyn, E. S.Andreeva, I. A. Nesterov, M. V. Klimenko, V. V. Klimenko, and Y. S. Tumanova (2016), Effects of ionizing energetic electrons and plasma transport in the ionosphere during the initial phase of the December 2006 magnetic storm, J. Geophys. Res. Space Physics, 121, 5880-5896, doi:10.1002/2016JA022622.

Line 97: (cm2 (cm2

Corrected

Lines 103-106: IT IS WRI comparative analysis at different phases of the solar cycle, we

have selected three couples of years: solar maximum (2001 and 2013), declining phase (2004 and

2016) and right before solar minimum (2007 and 2019).

I understand what is meant

For the comparative analysis at three different phases of the solar cycle, we have selected a

year from each solar cycle: solar maximum (2001 and 2013), declining phase (2004 and 2016)

and right before solar minimum (2007 and 2019)

Corrected

Line 107: that - with that with

Corrected

Lines 106-109: It is mentioned that in 2014 the temporal resolution of the acquired data changed and

the number of satellites increased. Could this increase in statistics also potentially affect the

observed effects in the two solar cycles?

The temporal resolution should not affect the location of ORB. It might affect the probability of observations of strong localized sporadic fluxes. The outer and inner ERB are large-scale structures. Hence, the increase of the temporal resolution should not affect the effects observed. In particular, we restrict our spatial resolution by cells 3° x 2° of geographical longitude and latitude, respectively. This should minimize the difference in spatial scales.

The number of satellites should not also affect seriously because we accumulate 1-year data for the comparison. As one can see from Table 1, the number of satellites was increasing up to 6 by the year of 2009.  After the year of 2014, the number was fixed at 5 probes on-orbit. Perhaps, the effects observed in the year 2001 might be affected by pure statistics of two satellites operating at local time 6-18. On the other hand, the strongest effects are found from the comparison of the years 2007 and 2009, which are provided by the best statistics.

In order to clarify this important issue, we modify Table 1, where we demonstrate the data availability for all the years considered.

It should be also noted, that the local time is provided for the equatorial region. At high latitudes, satellites cross practically all MLTs and, thus, for the 1-year data this effect can be also neglected. 

We have add an addition paragraph in order to discuss this very important issue:

“The increased number of observation platforms (from two to six by 2009), and the relatively large spatial scale of 3° x 2° however, has negligible effect on our study; since, except for 2001, the other years are well represented in the data and, thus, are directly comparable.”

Line 125: All these facts confirm the previous findings [1, 20, 21]. All these observations confirm

previous findings [1, 20, 21].

Corrected

For the first time such dynamics was discovered earlier in [20, 21].

to reformulate: dynamics was discovered earlier by [20, 21] Do they discuss

the accelerated shift of the north magnetic pole from Canada towards Siberia causing this effect?

This part was revised as the follows:

“This shift is more prominent in 2019, confirming earlier findings [1,20,21], and is related to the accelerated shift of the north magnetic dip pole from Canada towards Siberia [11]”

Lines 169-171: IT intensity of precipitations from the outer ERB

increased during the 24th cycle that might be explained by an increase of the solar wind

magnetosphere coupling during that cycle. ar cycle

when there are more high speed solar winds?

It might be. We modify the sentence accordingly:

 “An increased solar wind-magnetosphere coupling might provide an additional explanation, which in turn might be caused by a higher efficiency of interaction with high-speed solar winds, and requires further sophisticated investigations.”

Line 174: intensity > 104 --> intensity > 104

Corrected

Lines 184- However, the outer ERB demonstrates an increase of the electron fluxes as a possible source of the inner ERB. Please explain/reformulate (demonstrates?).

Revised as the following:

“However, although the outer ERB exhibits an increase of the electron fluxes, the inner ERB shows correspondingly decrease. This suggests that the outer ERB should not be considered as a source for the inner ERB (as has been claimed in the literature).”

Lines 209-211: IT IS WRITTEN We have conducted an analysis of continuous observations conducted

during the 23rd and 24th solar cycles (i.e., from 1998 to 2019) by low-orbit NOAA/POES polar

satellites.

Suggestion: aper we have presented an analysis performed on continuous

observations covering the 23rd and 24th solar cycles (i.e., from 1998 to 2019) recorded by

detectors onboard the low Earth orbit NOAA/POES polar satellites.

Revised

Line 223: due anomalous due to anomalous

Corrected

Line 227: Potentially mention future (ongoing) work (e.g., Line 172, other datasets) that could be

performed.

We add the following sentences:

“In order to separate the effects of geomagnetic activity and the geomagnetic field changes in the electron ERB dynamics, we plan additional sophisticated analysis with more data in the near future.”

Reviewer 3 Report

This is an original paper. I recommend publication in its current form.

Author Response

We thank the Reviewer#3 for his kind comment. We have improved the English.

(x) English language and style are fine/minor spell check required

This is an original paper. I recommend publication in its current form.

Round 2

Reviewer 1 Report

Line 4: in the LEO --> in LEO

Line 7: area of South Atlantic --> area of the South Atlantic

Line 11 and Line 242: The word "echelon" is used. Is this a specific term? Is the word singular or plural? Should it not be "altitudes"?

Line 15: with energy of tens --> with energies of ten

Line 47: magnetic dip-pole moving with an acceleration from --> magnetic dipole moving with an increased acceleration from

Line 58: Is there another word than "death"?

Lines 61-62: IT IS WRITTEN: "However, the mechanism of their transport from the ERB has not been fully investigated. Thus, energetic electrons at low latitudes are ..." The word "Thus" refers to what (which sentence?)?

Line 105: by zenith-directed detector. --> by a zenith-directed detector.

Line 140: dip-pole --> dipole

Line 142: However, lower latitudes exhibit effects a contrasting picture. --> However, at lower latitudes the observed effects exhibit a contrasting picture.

Line 177: What is meant by "outlay"? Should this not be "outline" or "present"? 

Author Response

We thank the reviewer for their very generous and detailed comments on the grammar in the manuscript. We have modified the manuscript as suggested (highlighted in blue in the revised manuscript).

Reviewer 2 Report

The authors answered my questions. I suggest that the manuscript be accepted for publication.

Author Response

We thank the reviewer for their patience, and significant questions that made us make our manuscript better.
